# A Plant-Based Food Guide Adapted for Low-Fat Diets: The VegPlate Low-Fat (VP_LF)

**DOI:** 10.3390/foods13244050

**Published:** 2024-12-15

**Authors:** Luciana Baroni, Gianluca Rizzo, Martina Zavoli, Maurizio Battino

**Affiliations:** 1Scientific Society for Vegetarian Nutrition—SSNV, Mestre, 30171 Venice, Italy; luciana.baroni@scienzavegetariana.it (L.B.); martina.zavoli@scienzavegetariana.it (M.Z.); 2Joint Laboratory on Food Science, Nutrition, and Intelligent Processing of Foods, Polytechnic University of Marche, Italy, Universidad Europea del Atlántico Spain and Jiangsu University, China, Via Pietro Ranieri 65, 60131 Ancona, Italy; m.a.battino@univpm.it; 3International Joint Research Laboratory of Intelligent Agriculture and Agri-Products Processing, Jiangsu University, Zhenjiang 212013, China; 4Department of Clinical Sciences, Polytechnic University of Marche, Via Pietro Ranieri 65, 60131 Ancona, Italy; 5Research Group on Foods, Nutritional Biochemistry and Health, Universidad Europea del Atlántico, Isabel Torres 21, 39011 Santander, Spain

**Keywords:** low-fat diet, vegan diet, food guide, VegPlate

## Abstract

Strong evidence supports the paramount importance of the composition of the diet for health. Not only diet should provide nutritional adequacy, but some foods and dietary components can also support the management of common chronic diseases, with mechanisms independent of nutritional adequacy. Among the various intervention diets, low-fat vegan diets have been shown to be effective for cardiometabolic health, mainly influencing insulin resistance, adiposity, and blood lipids. This type of diet relies on reducing or eliminating all added fats and choosing low-fat foods, mainly unprocessed whole-plant foods. We hereby propose a tool for planning low-fat vegan diets, the VegPlate Low-Fat (VP_LF), which has been obtained from a specific adaptation of the VegPlate method, which was already presented in previous publications for adults and some life stages and situations. The reduction in fats in the diet, which ranges between 10% and 15% of total energy, and the varied inclusion of foods from plant groups make it easier to provide adequate amounts of all nutrients with a normal- or lower-calorie intake, in comparison with diets that do not limit fat intakes. We expect that this new proposal will help nutrition professionals embrace low-fat diets as a first-line intervention for individuals affected by different health conditions who can benefit from these diets.

## 1. Introduction

Vegetarian diets include lacto-ovo-vegetarian and vegan diets. Both diets exclude the consumption of animal flesh, and the latter also excludes the consumption of animal derivatives such as eggs, dairy products (milk and cheese), and honey. In both subtypes, the main calorie sources come from the consumption of grains, legumes, fruits, vegetables, nuts and seeds, and plant oils. The nutritional adequacy of vegetarian diets has been described by the main international nutrition organizations [1,2,3,4].

Strong evidence supports the paramount importance of the composition of the diet for health. Not only should diet provide nutritional adequacy but also some foods and dietary components can support the management of common chronic diseases, with mechanisms independent of nutritional adequacy. Higher AHEI scores (Alternative Healthy Eating Index), which estimate diet quality, have been associated with a lower risk of chronic diseases including cardiovascular disease, cancer, and diabetes mellitus [5,6,7].

The extensive scientific literature supports the favorable health effect of vegetarian diets on cardiometabolic health and cancer [8,9,10,11,12]. Moreover, intervention trials with low-fat diets—mainly performed on overweight and/or diabetic adults—support the effectiveness of reducing the amount of fat in the diet for cardiometabolic health [13,14,15]. Despite that the global risk–benefit ratio of this kind of intervention cannot be established in the short duration of a trial, the paradigm shift proposed by Sabaté suggested that the risk of deficiency reported for some kinds of vegetarian diets does not overcome their favorable health effects [16]. The AHEI score has been shown to significantly improve on a low-fat vegan diet [17]. Compared with the NCEP diet (National Cholesterol Education Program), a low-fat vegan diet improved body weight, insulin sensitivity, the thermic effect of food, and resting metabolic rate after 14 weeks in 64 overweight, postmenopausal women [18].

This kind of diet relies on reducing or eliminating all added fats and choosing low-fat foods, mainly unprocessed whole-plant foods. To our knowledge, a universal method for planning low-fat vegan diets does not exist, and health professionals are puzzled when added fats in the diet are limited. Clinical trials with low-fat vegan diets are performed by offering dietary instructions to participants. Conversely, a method providing practical instructions for professionals, which can be used in their daily practice, could represent a useful tool. We hereby propose a food guide for planning low-fat vegan diets, the VegPlate Low-Fat (VP_LF), which has been obtained with a specific adaptation of the VegPlate method, already presented in previous publications for adults and some life stages and situations [19,20,21].

## 2. Materials and Methods

We developed a facilitative method for health professionals who choose to use a low-fat vegan diet as a non-pharmacological approach to the treatment of some cardiometabolic diseases: the VegPlate Low-Fat (VP_LF). To this aim, we referred to the VegPlate method, which was first published in 2018 to offer a simple food guide for well-planning vegetarian diets (lacto-ovo and vegan) [19].

### 2.1. The Basic VegPlate Method

In the basic VegPlate method, foods were classified into “food groups”, and a “servings system” was proposed for the amount of food to use. The food groups were represented in a main diagram in the form of a plate, subdivided into six areas: one area for each food group (grains, protein-rich foods, vegetables, fruits, nuts and seeds, and fats—the latter corresponding to added fats of plant origin), and an outer plate or glass for discretionary calories (i.e., the number of calories unnecessary to reach nutrient adequacy). The diagram also included 2 cross-sectional groups: calcium-rich foods and n-3-rich foods. In the center of the plate, vitamins B_12_ and D were collocated to highlight their importance in a well-planned diet. The only difference between the two subtypes of vegetarian patterns (lacto-ovo and vegan) was in the protein-rich food group, which included all the protein-rich foods of plant origin and, only for lacto-ovo-vegetarian, dairy and egg derivatives. The other five groups were identical. The food selection and serving size have been previously described [19]: for each food group, we selected the most representative plant foods from the Mediterranean tradition. The “servings system” suggested the amount of each food of the same food group providing similar amounts of energy and nutrients, making it possible to vary the food choices within the same group without the need for rigorous exchange lists. One serving of calcium-rich foods provides an average of 125 mg of calcium and should be included in the number of servings indicated in each group.

One serving of omega-3-rich foods provided an average of 2.5 g of alpha-linolenic acid (ALA). According to the serving system, for each group, the consumption of a defined number of servings is allowed to satisfy the calorie and nutrient requests.

Four versions of the VegPlate for specific life stages and situations have already been obtained by simply modifying the number of servings for each food group and adding supplementary small plates: Adult, Pregnancy and Lactation [19], Children and Adolescents [20], and Athlete [21]. Working with the serving system, i.e., determining the amount of food to consume for each group expressed as the number of servings, made it possible to satisfy the calorie and nutrient needs of each population group.

### 2.2. The Adaptation of the VegPlate to a Low-Fat Diet

A low-fat diet is a diet that minimizes the energy contribution from the macronutrient fat. To this end, it is necessary to choose low-fat foods and avoid added fats. Therefore, to obtain the VP_LF, we needed to modify the VegPlate main diagram: the two fatty food groups (nuts and seeds, and fats) were replaced by one group, the n-3-rich food group, to provide n-3 fats to the diet. Moreover, the protein-rich food group should only include foods of plant origin.

Therefore, the VP_LF is composed of 5 fundamental food groups: (a) grains; (b) plant protein-rich foods; (c) vegetables; (d) fruits; (e) n-3-rich foods, and a cross-sectional group of calcium-rich foods, formed by the richest foods in calcium present in all the food groups (with the exclusion of the n-3-rich food group). Vitamin B_12_ and vitamin D are always placed at the center of the plate, and discretionary calories are placed outside the plate, as a small dish or glass. The resulting diagram is presented in Figure 1. Since the number of servings varies according to the caloric intake, the areas are representative only of the food groups composing the plate, but not of the quantities.

Throughout the different VegPlates, the serving dimension is constant, and the same is true for the VP_LF, as shown in Table 1. The IEO (European Institute of Oncology) food database we used to calculate the nutritional composition of each serving was updated according to the 2022 release [22].

To reduce fat intake in the VP_LF, non-ultra-processed plant foods should be emphasized, and an n-3-fat supplement (at least 200–250 mg EPA/DHA) should be included to compensate for the reduction in the intake of ALA from 5 g/d, like in all the VegPlates, to 2.5 g/d. Moreover, discretionary calories should be derived from low-fat foods. It is also recommended to supplement vitamin B_12_ and vitamin D and to emphasize calcium-rich foods (see Table 2). To reduce fiber content and the mass of vegetables and fruits, it is suggested that they be consumed, at least in part, as extracts and juices.

## 3. Results

Like in all the different adaptations of the VegPlate, in the VP_LF, the average nutritional composition of one serving from each group was used to determine the daily number of servings to consume from each group to satisfy the Italian DRIs [23], which also met the USDA, Canada, and UK DRIs [24,25,26].

We choose to display the number of servings suggested for calorie requirements ranging from 1600 to 3000 kcal in Table 3 because, above this calorie level, nutrient needs are largely respected, and it is possible to obtain a higher calorie intake by simply adding more foods from the grains, protein-rich foods, vegetables, and fruits groups. We offer an example, which is not mandatory, for calorie intakes from 3100 to 4000 kcal in the Appendix A. According to FAO [27], n-3-fatty acids should represent 0.5–2% of total energy and should include 250 mg of EPA/DHA, which can also be endogenously converted from ALA. Two servings of n-3-rich foods from the VegPlate can satisfy these recommendations by providing the amount of ALA to be converted in EPA/DHA [28,29,30], thanks also to the low n-6/n-3 ratio of the menus (ranging from 1.63:1 to 1.98:1). Since above 3000 kcal, 2 servings of n-3-rich foods are planned to be consumed, EPA/DHA supplementation is unnecessary.

Table 4 shows the nutritional composition (energy, fiber, macronutrients, and main micronutrients) of the serving patterns proposed in Table 3, and the Italian, Canadian, UK, and US DRIs. As evident from Table 5, the average percentage of fats in the diets ranges from 15% (for the lower calories) to 10% (for the higher calories), and the n-6/n-3 ratio (calculated only from foods, not supplements) always falls far below the minimum 4:1 ratio recommended by the main institutions [31]. Appendix A shows the multiple steps needed to obtain a sample menu.

## 4. Discussion

Extensive scientific literature and several meta-analyses support the favorable health effect of vegetarian diets on cardiometabolic health and cancer [8]. In comparison with non-vegetarian diets, meta-analyses of observational [9,12,33] and intervention [34,35,36,37] studies aiming to evaluate the effects of vegetarian diets on body weight found they can favorably affect it. Vegetarian diets reduced diabetes risk in a meta-analysis of observational studies [38] and improved glycemic metabolism both in observational [12,33] and intervention studies [36,39,40]. In meta-analyses of observational studies, lower blood cholesterol levels were reported in vegetarians [9,12,33,41], and the same effect was found in meta-analyses of intervention studies [36,37,40,41]. A reduction in blood pressure was found in meta-analyses of observational studies [9,33,42] but was not consistently reported in meta-analyses of intervention studies [36,40,42]. The incidence and mortality for vascular disease, evaluated only in meta-analyses of observational studies, were reduced for ischemic heart disease [12,43,44,45] and cardiovascular disease [9,45]. Also, meta-analyses on cancer risk, which have been performed only on observational studies, found a reduction in the risk in vegetarians [12,43,46,47]. For all outcomes, when vegans were evaluated separately from other vegetarians, a further advantage was detected [9,12,35,41,47].

The abundance of fiber and water and the limited content of fats in unprocessed plant foods are responsible for their low-calorie density, reducing the risk of excessive energy intake. The thermic effect of foods increased after the consumption of large intakes of carbohydrates and low intakes of fats [48], and plant foods favorably influenced the regulation of appetite and the intake of food and energy by regulating the secretion of gastrointestinal hormones [49,50]. Providing the body with molecules with antioxidant properties, like compounds naturally contained in plants—especially when grown in harsh environments—can protect the body from the damage caused by oxidative stress [51]: the consumption of whole plant foods, rich in antioxidants, has been associated with a reduction in the risk of major chronic diseases [52]. Accordingly, it has been reported that a vegetarian diet can reduce the levels of oxidative stress markers and the oxidant–antioxidant balance compared to an omnivore diet [53,54]. Moreover, a plant-based diet can reduce iron accumulation in the tissues [55], which can be related to multiple health outcomes [56]. Plant foods can favor the growth of beneficial bacteria and a greater richness and diversity of gut microbiota: it has been reported that vegetarians have a higher *Prevotella*/*Bacteroides* ratio and a lower *Firmicutes*/*Bacteroidetes* ratio [57]. Short-chain fatty acids (acetate, propionate, and butyrate), produced during the bacterial fermentation of dietary fiber, elicit beneficial effects on immunity, inflammation, lipid and glucose metabolism, gut barrier, and blood–brain barrier integrity [58,59,60,61]. Plasma and urinary TMAO concentrations (trimethylamine N-oxide, a marker of cardiovascular disease risk) are increased by dietary protein, particularly of animal origin, and reduced in subjects following plant-based diets [62]. Low-grade chronic inflammation represents a common underlying factor in chronic diseases [63,64,65,66,67]. Lower levels of inflammatory biomarkers, mainly hsCRP, have been reported in vegetarians [68,69,70,71], suggesting that the reduction in their circulating levels in subjects following a plant-based diet could reduce the risk of chronic diseases. Accordingly, in a clinical trial performed on subjects affected by coronary artery disease, a −32% significant reduction in hsCRP (high-sensitivity C-Reactive Protein, a marker of risk for major adverse cardiovascular outcomes in coronary artery disease) was observed after 8 weeks of a vegan diet, in comparison with the diet proposed by the American Heart Association [72].

Adipose tissue and liver and muscle fat accumulation are associated with insulin resistance [73,74]. Moreover, it has been shown that dietary fat acutely increases insulin resistance in human skeletal muscles and glucose concentrations and insulin requirements in patients with type 1 diabetes [75,76] and that high-fat meals elicit negative effects on endothelial cells [77]. In contrast, intervention trials with low-fat diets, which have been performed mainly on overweight and/or diabetic adults, support the effectiveness of reducing fat content in the diet for cardiometabolic health: low-fat and vegan diets lowered the concentrations of intramyocellular and hepatocellular lipids, increased mitochondrial activity and postprandial metabolism, improved beta-cell function and insulin resistance, and favored glycemic control [13,14,15]. In a 74-week trial involving 99 type-2 diabetic subjects, in comparison with the diet of the American Diabetes Association, a low-fat vegan diet reduced body weight, HbA1c, total and LDL-cholesterol, and diabetic medication [78,79]. In a 2-year randomized trial comparing a low-fat, vegan diet with the National Cholesterol Education Program (NCEP) on 62 overweight, postmenopausal women, the low-fat vegan diet was associated with significantly greater weight loss than the NCEP diet at 1 and 2 years [80]. In a 16-week intervention trial involving 244 overweight subjects, a low-fat diet resulted in a reduction in body weight and hepatocellular and intramyocellular fat, and increased insulin sensitivity, compared with the habitual diet [15]. Compared with the Mediterranean diet, a low-fat vegan diet promoting ad libitum intake of unprocessed plant foods and the avoidance of added fat improved body weight, lipid concentrations, and insulin sensitivity, after 16 weeks in 62 overweight adults [81] and decreased dietary AGE intakes. In a randomized 16-week cross-over trial, changes in dietary AGEs correlated with changes in body weight, which decreased by 6 kg, compared with no change in the Mediterranean diet [82]. In a multicenter clinical trial performed on employees from 10 sites of a major US company with a body mass index ≥ 25 kg/m^2^ and/or a previous diagnosis of type 2 diabetes, a vegan low-fat diet improved body weight, plasma lipids, and, in individuals with diabetes, glycemic control after 18 weeks, compared with the habitual diet [83].

Roberts et al., in their 3-week intervention trials on adults and children with a low-fat, high-fiber diet combined with daily exercise, reported improvements in BP, oxidative stress, NO availability, inflammation, monocyte–endothelial interactions, and metabolic profile [84,85,86]. A 5-year intervention trial (the Lifestyle Heart Trial on Coronary Artery Disease patients) performed by Ornish et al. with a lifestyle-intensive program also included a vegetarian very low-fat diet (10% fat). In terms of coronary arteriography, the control group experienced a 27.7% relative worsening in the average percentage of stenosis, whereas the intervention group experienced a 7.9% relative improvement. Moreover, cardiac events occurred 2.5 times more frequently in the subjects who did not undergo the intervention [87].

In a 2-year intervention study on 93 patients affected by prostate cancer undergoing an intensive lifestyle program that included a vegan low-fat diet, only 5% of patients in the intervention group required traditional treatment, whereas 27% in the control group did [88]. Other authors reported a beneficial effect of a low-fat diet on prostate and breast cancer, often in association with exercise [89,90,91,92,93,94,95,96]. It was suggested that one of the mechanisms involved may be an increase in circulating IGFBP-1 (IGF binding protein-1) and a decrease in serum IGF-I, resulting in reduced cancer cell growth [92,93].

Moreover, it has been reported that a low-fat diet can mitigate the symptoms of menopause and the menstrual cycle by increasing the serum sex-hormone binding globulin concentration and influencing estrogen activity and AGE levels [97,98,99]. It can also reduce the intensity and frequency of migraine attacks [100,101].

A low-fat diet, included in a comprehensive lifestyle intervention, significantly increased telomerase activity in a pilot study, hypothetically slowing the aging process since telomere shortness in humans is emerging as a prognostic marker of disease risk, progression, and premature mortality [102].

Recently, it has been proposed that a whole-food, low-fat vegan eating pattern can be considered a new reference healthy eating pattern for a Universal Food Guide, which can serve as a template for sustainable and healthy national food guides [103].

We aimed to provide a practical tool for well-planned low-fat diets. An innovative plant-based food guide for vegetarians, the VegPlate, was conceived in 2017 and was proposed for adults, pregnant and breastfeeding women [19], children and adolescents [20], and athletes [21], thanks to adaptations of the basic structure of the VegPlate for adults. Using this method, we conceived the VP_LF, a plant-based food guide for low-fat diets: a reduction in added fats in the diet and the varied inclusion of foods from the plant groups, mainly unprocessed, allowed us to obtain a low-fat vegan diet, offering a calorie intake from fats ranging between 10% and 15% of the total energy. Natural plant foods do not contain cholesterol, and fatty acids are mostly polyunsaturated or monounsaturated (except for tropical fruits). Moreover, plant foods do not contain EPA and DHA, whose—non univocally—suggested average daily amount is 250 mg/d [23,104]. In the VP_LF guide, the only fat foods in the diet are placed in small amounts in the n-3-rich food group, which includes selected foods containing high amounts of alfa-linolenic acid (ALA). The proposed adequate intakes are 1.1 g/d for females and 1.6 g/d for males [32] or 0.5–2% of the total energy (corresponding to 8–60 kcal from n-3 in the range 1600–3000 kcal), including 250 mg EPA-DHA [23,27]. In the range of 1800–3000 kcal, VP_LF provides 3.76 to 4.10 g of ALA (34–37 kcal). It has been reported that the rates of endogenous conversion from ALA to EPA vary between 0.2% and 21%, and those to DHA vary between 0% and 9%, being higher in women [30]. The conversion efficiency can be enhanced by reducing enzymatic competition with linoleic acid, which essentially consists of increasing the intake of ALA, consuming monounsaturated fats in place of n-6 polyunsaturated fats, and maintaining a low n-6/n-3 ratio, for which the main institutions suggest a minimum ratio of 4:1 [31,105]. In the VP_LF guide, the n-6/n-3 ratio (calculated only from foods, not supplements) always falls far below the minimum 4:1 ratio but to safely respect the reference intake, the consumption of a DHA supplement is suggested (250 mg/d).

Currently, a widely accepted method to track PUFAs’ intake sufficiency is lacking. A commonly used marker is the Omega-3 index, which is thought to predict dietetic intakes of omega-3 polyunsaturated fatty acids through EPA and DHA quantification in red blood cell membranes [106]. This biomarker has been proposed to identify high-risk patients for cardiovascular disease and other neurocognitive illnesses [106,107]. Future research is needed to validate the reliability of the VP_LF diet and confirm the adequacy of dietary essential fatty acids, given their crucial role in chronic diseases.

The VP_LF guide includes only plant proteins, whose proposed amounts respect the Italian and US reference intakes [23,24]. Due to their lower digestibility and peculiar amino acid composition in plant foods, dietary protein adequacy in plant-based diets has long been debated [108,109] but has been approved by leading experts in the field [1,2,108,110,111]. Despite it being reported that vegetarian diets typically contain adequate amounts of protein, including adequate amounts of all 20 amino acids and, specifically, all the essential amino acids [111], adjustments for the bioavailability of protein with a 10–15% increase in intake have been applied [2,19,105]. Plant proteins are associated with healthy aging, a reduction in the risk of frailty in elderly individuals, and lower mortality [112,113,114], and their consumption can reduce the risk of obesity and other chronic diseases [115,116,117]. Moreover, plant protein can favorably influence gut health by regulating the microbiota composition, promoting a diverse and healthy population of gut bacteria [118]. VP_LF also addresses the modernization of the concept of “protein quality” from the perspective of planetary health [115], which favors plant protein since the protein conversion rate from plants to animals ranges from 20:1 to 4:1 (with an average of 9:1) [119].

The limitation of % total energy from fats to 10–15% results in the intake of more nutrient-dense calories, i.e., richer in macro and micronutrients, in comparison to diets that do not limit fats. Practically, limiting fats allows the intake of higher amounts of nutrients (different from fats) with the same calorie intake. Accordingly, the contents of all micronutrients, except for vitamin B_12_ and vitamin D, which are classified as critical nutrients in all plant-based diets, are shown in Table 4.

Vitamins B_12_ and D are placed at the center of the plate to emphasize their fundamental presence in a well-planned diet. Vitamin B_12_ (cobalamin) is a unique vitamin because it is synthesized by bacteria located in the environment and digestive systems of animals and is almost completely absent in plant foods unless not fortified. An adequate intake of 4 mcg/d has been proposed for adults [23,120], which is impossible to achieve with a plant-based diet since animal foods are the only source of this vitamin. Nevertheless, cobalamin malabsorption is a cross-sectional problem for all kinds of diets and can also compromise B_12_ status in people consuming animal foods [121,122]. Therefore, although cobalamin deficiency is common worldwide, it is well recognized that vegetarians, especially in the general population following plant-based diets, should supplement it. This issue is even more relevant in the elderly population, where the risk of B12 deficiency exists regardless of their overall dietary pattern [1,2,19,123]. If the latitude is favorable, vitamin D is produced in the skin by sunlight and then further processed in the liver and the kidney to obtain its active form. It is contained in fatty animal foods, so vegetarians are considered at-risk and are advised to pay attention to vitamin D status [19,123]. Nevertheless, it has been reported that vitamin D status is influenced mostly by supplementation, the degree of skin pigmentation, and the amount and intensity of sun exposure rather than by diet [124], so vitamin D deficiency appears to be a cross-sectional situation and has been compared to a pandemic [125].

In the VegPlate method, the group of foods rich in calcium has been conceived as a cross-sectional group, and Table 4 shows that calcium needs are satisfied with a varied choice of plant foods. In any case, the knowledge of calcium-rich plant foods can help nutrition professionals plan a diet that respects calcium requests. This problem is really only applicable to very-low-calorie intakes, but in this situation, it should be remembered that mineral water can provide highly absorbable calcium in moderate amounts, with no energy [2].

Finally, in his review, Storz reported that adherence to a low-fat vegan diet in type-2 diabetic subjects was greater than adherence to conventional diets in several studies since more than 50% of individuals met the criteria for high adherence in most studies, suggesting that physicians should advocate for this diet more frequently [126].

In summary, the theoretical composition calculated on ideal intakes, together with particular attention to the intake of critical nutrients, show that in the VP_LF, nutrition adequacy is achieved.

## 5. Conclusions

The benefits of a low-fat diet combined with a vegan plan have been explored in clinical trials. This promising approach appears to be beneficial to health, is consistent with the planetary goals of reducing the environmental impact of diets, and, with good and regular support, shows solid adherence rates to low-fat vegan diets. Nevertheless, to date, no food guide for planning low-fat vegan diets is available to health professionals. The VegPlate_Low-Fat (VP_LF) aims to present a new proposal for an easy-to-use vegetarian food guide for the planning of low-fat vegan diets to be used by health professionals in their daily activities. Despite that further research to confirm and eventually enhance the application of low-fat vegan diets in clinical practice is warranted, we hope that this new proposal can represent a useful tool for nutrition professionals and can be considered a complementary first-line, inexpensive approach to cardiometabolic diseases. We expect that further intervention studies performed with real diets based on the VP_LF method can confirm its effectiveness for combining nutrition adequacy and therapeutic advantages. It would also be useful to compare their effectiveness with that of omnivorous diets with similar amounts of fats.

## Figures and Tables

**Figure 1 foods-13-04050-f001:**
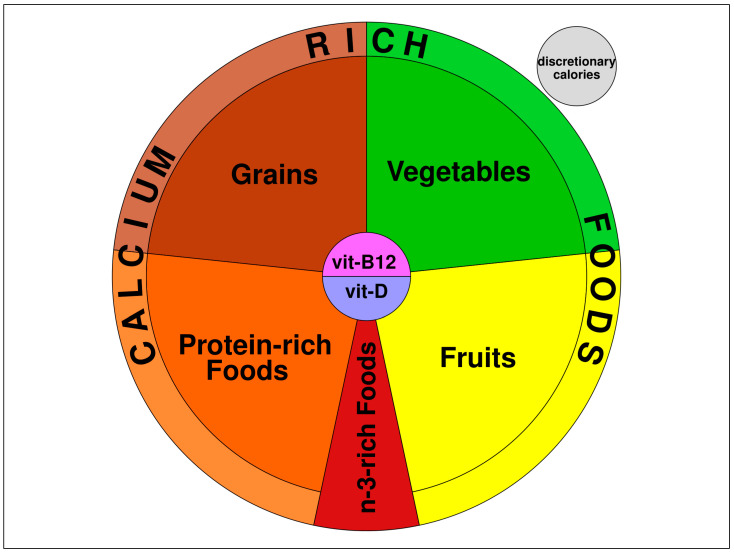
The VegPlate Low-Fat (VP_LF).

**Table 1 foods-13-04050-t001:** The serving size of the foods composing the VP_LF [22].

Food	Serving Size
1. Grains	
Bread, baked cereals (dried)	30 g
Grain cereals, bulgur, couscous	30 g
Pasta (dried)	30 g
Pop-corn (cooked)	30 g
Ready-to-eat cereals	30 g
Non-dairy milk from cereals	200 mL
(Potatoes—if consumed frequently)	120 g
2. Protein-rich foods	
Legumes (dried)	30 g
Tofu or tempeh	80 g
Meat alternatives	30 g
Non-dairy milk from soy	200 mL
Soy yogurt	125 mL
3. Vegetables	
Raw or cooked vegetables	100 g
Vegetable juice	100 mL
4. Fruits	
Raw fruits	150 g
Cooked fruits	150 g
Fruit juice	150 mL
Dried fruits	30 g
5. n-3-rich foods	
Flaxseeds (grounded)	10 g
Flaxseed oil	5 g
Walnuts	30 g (n = 6)
Chia seeds (ground).	15 g
6. Calcium-rich foods	
Listed in Table 2	

**Table 2 foods-13-04050-t002:** The calcium-rich foods of the VP_LF [22].

Food	Ca (mg/100 g)	Serving Size	Ca (mg/Serving)
Grains			
Non-dairy milk from rice (enriched with calcium)	123.5	200 mL	247 (=2 servings)
Protein-rich foods			
Soy yogurt (enriched with calcium)	132	125 mL	165
Non-dairy milk from soy (enriched with calcium)	120	200 mL	240 (=2 servings)
Tempeh	120	80 g	96
Tofu	105	80 g	84
Vegetables			
Dandelion	187	100 g	187
Watercress	170	100 g	170
Rocket	160	100 g	160
Chicory	150	100 g	150
Garden cress	131	100 g	131
Green radicchio	115	100 g	115
Turnip tops	97	100 g	97
Cardoon	96	100 g	96
Endive	93	100 g	93
Artichoke	86	100 g	86
Broccoli	72	100 g	72
Fruits			
Figs (dried)	280	30 g	84
Water			
Mineral water, calcium 350 mg/L	35	350 mL	125
Tap water, calcium 100 mg/L	10	1250 mL	125

**Table 3 foods-13-04050-t003:** Number of servings suggested for calorie intake from 1600 to 3000 kcal.

	Grains	Protein-Rich Foods	Vegetables	Fruits	n-3-Rich Foods	Discretionary Calories
1600	8	4	8	2	1	141
1700	9	4	8	2	1	160
1800	9	4	8	3	1	191
1900	10	4	8	3	1	211
2000	10	4	8	4	1	242
2100	11	4	9	4	1	238
2200	11	4	9	5	1	269
2300	12	4	9	5	1	289
2400	12	4	9	6	1	320
2500	13	4	9	6	1	339
2600	13	4	9	7	1	370
2700	14	4	9	7	1	389
2800	14	4	9	8	1	420
2900	15	4	9	8	1	439
3000	15	4	9	9	1	471

**Table 4 foods-13-04050-t004:** Nutritional composition of the serving patterns proposed in Table 3 ([23,25,26,32]).

	Protein (g)	Carbo Hydrate (g)	Fat (g)	Fiber (g)	Iron (mg)	Calcium (mg)	Zinc (mg)	Vitamin B1 (mg)	Vitamin B2 (mg)	Vitamin B3 (mg)	Folate (mcg)	ALA (g)
1600	75.25	228.93	27.27	50.35	25.21	1087.72	11.41	1.90	2.38	22.30	947.48	3.76
1700	77.72	245.47	27.90	51.91	25.88	1119.67	11.87	1.98	2.43	23.62	961.21	3.77
1800	78.77	260.66	28.21	54.48	26.52	1154.10	12.09	2.04	2.49	24.17	975.36	3.80
1900	81.24	277.20	28.84	56.04	27.19	1186.06	12.55	2.11	2.54	25.49	989.08	3.81
2000	82.29	292.39	29.14	58.61	27.83	1220.49	12.77	2.17	2.60	26.04	1003.23	3.84
2100	86.74	312.16	30.07	62.46	29.78	1308.22	13.63	2.31	2.80	28.17	1087.09	3.91
2200	87.79	327.36	30.37	65.03	30.42	1342.65	13.85	2.37	2.86	28.72	1101.24	3.94
2300	90.26	343.89	31.00	66.59	31.09	1374.60	14.31	2.45	2.91	30.04	1114.96	3.95
2400	91.31	359.09	31.31	69.16	31.73	1409.04	14.53	2.51	2.97	30.59	1129.11	3.98
2500	93.78	375.62	31.94	70.72	32.40	1440.99	14.98	2.58	3.03	31.91	1142.83	3.99
2600	94.83	390.82	32.24	73.29	33.04	1475.43	15.20	2.64	3.08	32.46	1156.98	4.02
2700	97.30	407.35	32.87	74.86	33.71	1507.38	15.66	2.72	3.14	33.78	1170.70	4.03
2800	98.35	422.55	33.18	77.42	34.35	1541.81	15.88	2.78	3.19	34.33	1184.85	4.06
2900	100.82	439.08	33.81	78.99	35.01	1573.77	16.34	2.85	3.25	35.65	1198.58	4.07
3000	101.87	454.28	34.11	81.55	35.66	1608.20	16.56	2.91	3.31	36.20	1212.73	4.10
Italian LARN [23]	54–63 (0.9 g/kg/d)	45–60% totalEn	20–35% totalEn	12.6–16.7 g/1000 kcal/d	10–18	950–1100	9–12	0.4	1.6	18	330	0.5–2% totalEn
US DRIs [32]	46–56	130	nd	21–38	8–18	1000–1200	8–11	1.1–1.2	1.1–1.3	14–16	400	1.1–1.6
UK DRVs [25]	46.5–53.3 (0.75 g/kg/d)	39%	35%	12–24	8.7–14.8	700	7–9.5	0.4 mg/1000 kcal	1.1–1.3	6.6 mg/1000 kcal	200	0.2% totalEn
Canada DRIs [26]	0.8 g/kg/d	130	nd	21–30	8–18	1000–1200	8–11	1.1–1.2	1.1–1.3	14–16	400	1.1–1.6

**Table 5 foods-13-04050-t005:** Percentage of total energy from macronutrients and discretionary calories, and amount of LA, ALA, and n6/n3 ratio of the serving patterns proposed in Table 3.

	Protein	Carbohydrate	Fat	Discr. Calories	LA (g)	ALA (g)	n-6/n-3 Ratio
1600	19%	57%	15%	9%	6.13	3.76	1.63
1700	18%	58%	15%	9%	6.31	3.77	1.67
1800	18%	58%	14%	11%	6.40	3.80	1.69
1900	17%	58%	14%	11%	6.58	3.81	1.73
2000	16%	58%	13%	12%	6.67	3.84	1.74
2100	17%	59%	13%	11%	6.92	3.91	1.77
2200	16%	60%	12%	12%	7.01	3.94	1.78
2300	16%	60%	12%	13%	7.19	3.95	1.82
2400	15%	60%	12%	13%	7.28	3.98	1.83
2500	15%	60%	11%	14%	7.46	3.99	1.87
2600	15%	60%	11%	14%	7.55	4.02	1.88
2700	14%	60%	11%	14%	7.73	4.03	1.92
2800	14%	60%	11%	15%	7.82	4.06	1.93
2900	14%	61%	10%	15%	8.00	4.07	1.97
3000	14%	61%	10%	16%	8.09	4.10	1.98

## Data Availability

The original contributions presented in the study are included in the article and Appendix A, and further inquiries can be directed to the corresponding author.

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
