# Peer review of "A Plant-Based Food Guide Adapted for Low-Fat Diets: The VegPlate Low-Fat (VP_LF)"

_foods, 2024, doi:10.3390/foods13244050_

Round 1

Reviewer 1 Report

Comments and Suggestions for Authors

In the attachment. 

Author Response

Rev#1

Congratulations on your insights regarding the proposed VegPlate low-fat guide.

I recommend some minor enhancements to your work. Additionally, consider using more

scientific terminology in certain places and improving readability and articulation in

some places (for both, I only wrote about it in a few places, but there are more in the introduction and discussion sections).

It would be helpful to avoid ambiguous terms such as "help" and "fats" in order to provide more straightforward explanations. Strengthening and balancing some of your claims with references would also be beneficial.

Best of luck with the publishing process!

We thank Rev#1 very much for his/her appreciation and valuable suggestion. We tried to make some modifications to obtain a clear distinction between “fat” as macronutrients and fatty foods, and added more references.

Abstract:

The abstract is clear, concise, and provides enough information for the reader.

- ‘Help’ or rather support?

- That do not limits ‘fat’ or rather fat intake?

-‘We hope that this new proposal' or rather ‘We expect that ‘

Thank you for the suggestion, we modified it accordingly

Introduction:

Line 39: Evaluate whether the Italian association is the leading international one or consider

including a few more reputable international nutrition associations to add weight to the phrase"main".

Thank you for the suggestion. We added the following references:

-American Dietetic Association; Dietitians of Canada Position of the American Dietetic Association and Dietitians of Canada: Vegetarian Diets. J Am Diet Assoc 2003, 103, 748–765, doi:10.1053/jada.2003.50142.

-National Health Services. The Vegetarian Diet. Available online: https://www.nhs.uk/live-well/eat-well/how-to-eat-a-balanced-diet/the-vegetarian-diet/ (accessed on 2 December 2024).

Line 47: Include recent significant meta-analyses and umbrella reviews alongside self-citation to enhance objectivity.

Thank you for the suggestion. We added the following references:

-Landry, M.J.; Senkus, K.E.; Mangels, A.R.; Guest, N.S.; Pawlak, R.; Raj, S.; Handu, D.; Rozga, M. Vegetarian Dietary Patterns and Cardiovascular Risk Factors and Disease Prevention: An Umbrella Review of Systematic Reviews. Am J Prev Cardiol 2024, 20, 100868, doi:10.1016/j.ajpc.2024.100868.

-Capodici, A.; Mocciaro, G.; Gori, D.; Landry, M.J.; Masini, A.; Sanmarchi, F.; Fiore, M.; Coa, A.A.; Castagna, G.; Gardner, C.D.; et al. Cardiovascular Health and Cancer Risk Associated with Plant Based Diets: An Umbrella Review. PLoS One 2024, 19, e0300711, doi:10.1371/journal.pone.0300711.

-Oussalah, A.; Levy, J.; Berthezène, C.; Alpers, D.H.; Guéant, J.-L. Health Outcomes Associated with Vegetarian Diets: An Umbrella Review of Systematic Reviews and Meta-Analyses. Clin Nutr. 2020, 39, 3283–3307, doi:10.1016/j.clnu.2020.02.037.

-Dinu, M.; Abbate, R.; Gensini, G.F.; Casini, A.; Sofi, F. Vegetarian, Vegan Diets and Multiple Health Outcomes: A Systematic Review with Meta-Analysis of Observational Studies. Critical Reviews in Food Science and Nutrition 2017, 57, 3640–3649, doi:10.1080/10408398.2016.1138447.

Line 50: While the study produced several positive health results, it is important to highlight

that there was still micronutrient insufficiency regarding vitamins B12 and D and calcium, zinc, potassium, and possibly selenium while exceeding sodium intake. Balance the results obtained

We appreciate this comment very much, although evaluating the risk-benefit ratio of this intervention is beyond the scope of our paper. To clarify this concept, we added the following sentence (lines 53-57):

Despite the global risk-benefit ratio of this kind of intervention can not be established in the short time of a trial, the paradigm shift proposed by Sabaté et al. suggested that the risk of deficiency reported for some kind of vegetarian diets does not overcome their favorable health effects [16].

Line 71 (and 99): If you are referring to added fats, remember that plant foods also contain fats. Therefore, specify "added fats."

Sure, this is the reason why it is “low-fat” and not “zero-fat”!

In the VegPlate only foods are indicated, so the term “Fats” comprises only foods (not the macronutrient), in the form of  “added Fats”

To frame this concept, we added (lines 82-83):

fats – the latter corresponding to added fats of plant origin)

Line 92: Consider whether 'doable' would be more appropriate than 'easy.'

Thank you, we corrected

Line 105: Should we also consider including EPA/DHA omega-3 in the middle, given the

insufficient conversion from ALA, even in low-fat diets? At least elaborate your decision.

No, because over 3000 kcal they are unnecessary (see later)

Table 1: Please make sure to indicate whether the products (cereals, yoghurt, milk) are fortified or non-fortified also for Table 1.

The market offers the same food in non-fortified and fortified form. In Table 1 we indicated the food serving offering similar amounts of energy and macronutrients. So, we detailed calcium-rich food separately, in Table 2.

Line 121-122: It is claimed that EPA/DHA supplementation is only needed if you reduce your

ALA intake from 5 grams to 2.5 grams and not in other cases. Add valid citations to support it.

We thank reviewer#1 for the request. We already coped with this issue in the discussion.

FAO/WHO (ref 27) and EFSA 2010 (Scientific Opinion on Dietary Reference Values for fats, including saturated fatty acids, polyunsaturated fatty acids, monounsaturated fatty acids, trans fatty acids, and cholesterol. The EFSA J 2010;8:1461-1568), recommends the intake of 0.5-2% total energy from omega-3, which should include 0.25 g in the form of EPA/DHA. For the diet with the highest calorie content (=/> 3100 kcal) this means that at least 1.55 g of omega-3 should be consumed, 1.52 g in the form of ALA and 0.25 g in the form of EPA/ DHA. Considering that the consumption of 2 servings of omega-3 rich foods provides on average 5.0 g of ALA (according to the USDA database, the ALA average content of 1 serving is 2.57 g: 2.7 g for flaxseed oil, 2.3 g for flaxseeds, 2.7 g for walnuts), the remaining 3.45 g of ALA can be converted in 0.276 g of EPA in adult males (conversion rates about 8%) and 0.31 g DHA in adult females (conversion rates about 9%)( ref 30). This estimated conversion rate can be even higher when considering that the VegPlate eliminates other added fats, which minimizes the omega-6:omega-3 ratio, which in turn maximizes the efficiency of the conversion pathway (ref. 28,29). The omega-6:omega-3 ratio in the different menus ranges from 1.63:1 to 1.98:1.

We completed the previous sentence as follows (lines 166-184):

According to FAO [27], n-3-fatty acids should represent 0.5-2% of total energy and should include 250 mg of EPA/DHA, which can also be endogenously converted from ALA. Two servings of 
n-3-rich foods from the VegPlate can satisfy these recommendations by providing the amount of ALA to be converted in EPA/DHA [28–30], thanks also to the low n-6:n-3 ratio of the menus (ranging from 1.63:1 to 1.98:1).

Line 124: You are generally very precise in determining entries, enhancing this for the intake

of vitamins B12 and D, too. Please specify the databases used to assess the nutritional composition of selected food groups (add the reference). Also, indicate whether heat exchangers were considered when preparing the dish.

The used database is reported in reference 22 and refers to raw foods

We moved to line 132 the following sentence

The IEO (European Institute of Oncology) food database we used to calculate the nutritional composition of each serving was updated according to the 2022 release [22].

And added the reference number to Tables 1 and 2.

Discussion:

In this discussion, you primarily focus on the health outcomes of several well-known vegan

studies. However, your main aim is to promote a well-designed low-fat diet, ensuring that such a diet is nutritionally sufficient for everyone.

Add a discussion on the existence of nutritional adequacy and the challenges of these studies

(also compared with omnivorous diets), and appropriately integrate this into your results and

contributions.

As above discussed, evaluating the risk-benefit ratio of this intervention is beyond the scope of our paper. To clarify this concept, we added the following sentence (lines 53-57):

Despite the global risk-benefit ratio of this kind of intervention can not be established in the short time of a trial, the paradigm shift proposed by Sabaté suggested that the risk of deficiency reported for some kinds of vegetarian diets does not overcome their favorable health effects [16].

Moreover, to address Rev#1 suggestion, we added the following sentences

(lines 319):

We aimed to provide a practical tool for well-planning low-fat diets.

(Lines 401-403)

In summary, the theoretical composition, calculated on ideal intakes, together with the respect of the particular attention for the intake of critical nutrients, show that in the VP_LF nutrition adequacy is respected.

And (lines 412-414)

We expect that further intervention studies performed with real diets based on the VP_LF method can confirm its effectiveness for joining nutrition adequacy with therapeutic advantages. It would also be useful to compare their effectiveness with that of omnivorous diet with similar amounts of fats.

Line 239: While the mechanistic explanation has been provided, it is important to exercise

caution until studies support this proposed plate. In addition, add a reference and discuss the

relevance of the omega-3 index variable, specifically whether a dosage of 250 mg of EPA/DHA

achieves levels above 4%.

We agree with Rev#1 about the crucial role of trials and other studies in confirming a theoretic method. Unfortunately, only clinical data can validate our proposal, even if there are strong clues from available literature about its reliability. To expand on this, we added the following part (at line 348-354). We thank Rev#1 for their valuable suggestion.

Currently, a widely accepted method to track the PUFAs' intake sufficiency is lacking. A commonly used marker is the Omega-3 index which is thought to predict dietetic intakes of omega-3 polyunsaturated fatty acids through EPA and DHA quantification in red blood cell membranes [106]. This biomarker has been proposed to identify high-risk patients for cardiovascular disease and other neurocognitive illnesses [107,108]. Future research is needed to validate the reliability of the VP_LF diet and confirm the adequacy of dietary essential fatty acids, given their crucial role in chronic diseases.

Line 243: Please replace the period with a comma.

We thank very much Rev#1 for the correction of this typo

Line 240-251: Maybe add a sentence or two about the dilemma or myth regarding protein

quality before discussing bioavailability.

We thank very much Rev#1 for the suggestion. We added the following sentence (lines 356-359)

Due to their lower digestibility and peculiar amino acid composition in plant foods, dietary protein adequacy in plant-based diets has been a long time debated [109,110], but has been approved by leading experts in the field [1,2,109,111,112].

Lines 253-254: You may be correcting the readability or comprehensibility of the sentence.

We thank Rev#1 very much. We rephrased the sentence as follows (lines 370-373):

The limitation of % total energy from fats to 10-15% results in the intake of more nutrient-dense calories, i.e. richer in macro and micronutrients, in comparison to diets that do not limit fats. Practically, limiting fats allows the intake of higher amounts of nutrients (different from fats) with the same calorie intake.

Lines 252-258: Combine several paragraphs, especially since they are related.

Thank you, we made a single paragraph discussing vitamin B12 and D

Line 266: Suggested to add: This issue is even more relevant in the elderly population, where

the risk of B12 deficiency exists regardless of their overall dietary pattern.

Thank you very much, it is a very important specification, we added the sentence suggested.

Conclusion:

Please provide further study directions that would enhance the scientific validity of your

VegPlate low-fat proposal, such as testing it in an actual intervention and comparing it with a

well-designed low-fat omnivorous diet.

We thank rev#1 very much for the suggestion. We added the following sentence (lines 424-429)

We expect that further intervention studies performed with real diets based on the VP_LF method can confirm its effectiveness for joining nutrition adequacy with therapeutic advantages. It would also be useful to compare their effectiveness with that of omnivorous diet with similar amounts of fats.

Reviewer 2 Report

Comments and Suggestions for Authors

Before this, the authors provided a Mediterranean-based food guide for Italian adult, pregnant, and lactating vegetarians, and published tha article entitled “VegPlate: A Mediterranean-Based Food Guide for Italian Adult, Pregnant, and Lactating vegetarians”. Here, the authors provided another useful diet guide for vegans. The manuscript is very interesting, well-written, and well-organized. However, some issues should be improved.

How to match the food composition and and calories you listed in Tables, and how to choose the food according to the intake calories.

How to understand Calcium-rich foods mentioned in the context

Editing errors:

Please check the table format

Insert the figure title below the figures

Comments on the Quality of English Language

English is fine

Author Response

Before this, the authors provided a Mediterranean-based food guide for Italian adult, pregnant, and lactating vegetarians, and published tha article entitled “VegPlate: A Mediterranean-Based Food Guide for Italian Adult, Pregnant, and Lactating vegetarians”. Here, the authors provided another useful diet guide for vegans. The manuscript is very interesting, well-written, and well-organized. However, some issues should be improved.

The Authors are grateful to Rev#2 for the words of appreciation and the valuable suggestion

How to match the food composition and and calories you listed in Tables, and how to choose the food according to the intake calories.

We thank Rev#2 very much for the criticism. In order to clarify the process, we added Table 2S, and the following sentence (Lines 194-195)

Table 2S shows the multiple steps needed to obtain a sample menu.

How to understand Calcium-rich foods mentioned in the context

We thank Rev#2 for pointing out this uncertainty. We provided more details in lines 95-96

One serving of calcium-rich foods provides an average of 125 mg of calcium and should be included in the number of servings indicated in each group.

Editing errors:

Please check the table format

Thank you very much. We reformatted Tables 1 and 2

Insert the figure title below the figures

We thank Rev#2, we corrected

Reviewer 3 Report

Comments and Suggestions for Authors

1. When describing the relationship between vegetarian diets and health, the authors are encouraged to compare in greater detail the differences in nutrient composition and health effects across various types of vegetarian diets. Additionally, for studies on the relationship between chronic diseases and diet, a discussion on the differences in epidemiological studies across populations from different regions could be included.

2. What are the advantages of the VegPlate Low-Fat (VP_LF) model in existing low-fat diet planning? Please highlight these advantages.

3. It is suggested to include practical application cases, a specific food categorization system, and a portion size framework to better illustrate the application process and its value in introducing the basic VegPlate methodology,

4. Why the authors selected specific food substitutions or adjustments in the transition from VegPlate to VP_LF, and why nuts and seeds were chosen to be combined with fats in an n-3 rich food group, rather than exploring other combinations or screening and definition approaches.

5. In the selection of populations, in addition to comparisons with Italian and US DRIs, it may be valuable to add comparisons with the nutritional standards of other countries or international organizations.

6. In discussing the effects of low-fat vegan diets on various diseases, the authors could explore the interconnections between the mechanisms of action in greater depth, rather than merely listing the results of individual studies.

7. It is recommended that references be primarily from the last five years.Please carefully format references according to the formatting requirements of the Foods, many articles are missing page numbers. 

Author Response

We thank very much Rev#3 for his/her valuable suggestion. We tried to improve the manuscript’s contents according to Rev#3 inputs, with the related references.

1.When describing the relationship between vegetarian diets and health, the authors are encouraged to compare in greater detail the differences in nutrient composition and health effects across various types of vegetarian diets. Additionally, for studies on the relationship between chronic diseases and diet, a discussion on the differences in epidemiological studies across populations from different regions could be included.

We appreciate Rev#3’s suggestion, even if the issues regarding nutrient adequacy is addressed in different sections of the paper and the primary aim of the paper is to propose a Food Guide for well-planning low-fat vegan diet, and not to review literature on nutrition adequacy of vegetarian diets and their relationship with health, which would require a dedicate publication. Nevertheless, to address Rev#2’s request, we added the following sentences (lines 209-223):

In comparison with non-vegetarian diets, meta-analyses of observational [9,12,33] and intervention [34–37] studies aiming to evaluate the effects of vegetarian diets on body weight found they can favorably affect it. Vegetarian diets reduced diabetes risk in a meta-analysis of observational studies [38], and improved glycemic metabolism both in observational [12,33] and intervention studies [36,39,40]. In meta-analyses of observational studies lower blood cholesterol levels were reported in vegetarians [9,12,33,41], and the same effect was found in meta-analyses of intervention studies [36,37,40,41]. A reduction of blood pressure was found in meta-analyses of observational studies [9,33,42], but was not consistently reported in meta-analyses of intervention studies [36,40,42]. The incidence and mortality for vascular disease, evaluated only in meta-analyses of observational studies, were reduced for ischemic heart disease [12,43–45] and cardiovascular disease [9,45]. Also meta-analyses on cancer risk, which have been performed only on observational studies, found a reduction of the risk in vegetarians [12,43,46,47]. For all the outcomes, when vegans have been evaluated separately from other vegetarians, a further advantage was detected [9,12,35,41,47].

  1. What are the advantages of the VegPlate Low-Fat (VP_LF) model in existing low-fat diet planning? Please highlight these advantages.

We thank very much Rev#3 for pointing out this aspect. We added the following sentence (lines 62-67).

To our knowledge, a universal method for planning low-fat vegan diets does not exist, and health professional can be puzzled when added fats in the diet are limited. Actually, clinical trials with low-fat vegan diets are performed by offering dietary instruction to participants. Conversely, a method providing practical instruction for professionals, which can be used in their daily practice, could represent a useful tool.

  1. It is suggested to include practical application cases, a specific food categorization system, and a portion size framework to better illustrate the application process and its value in introducing the basic VegPlate methodology,

We thank Rev#3 for this suggestion.

We added table 2S, proposing a suggested menu, and the following sentence (Lines 194-195)

Table 2S shows the multiple steps needed to obtain a sample menu.

  1. Why the authors selected specific food substitutions or adjustments in the transition from VegPlate to VP_LF, and why nuts and seeds were chosen to be combined with fats in an n-3 rich food group, rather than exploring other combinations or screening and definition approaches.

We thank Rev#3 for this criticism.

Since the VegPlate method has already been published for other lifestages and situation, and the method described in detail elsewhere, we’d avoid doing a rendition of previous publications and summarized in this paper the main concepts supporting the method. Actually, on the basis of the existing literature on low-fat vegan diets, we excluded animal and non-processed plant foods (from the protein-rich-foods group) and limited the consumption of nuts & seeds and added oils. In comparison with the basic VegPlate, maintaining these two food groups in the diagram had no practical sense, since they participate to the diet only for contributing to n-3 fatty acid intakes: for this reason, they have been grouped into the n-3-rich foods group.

  1. In the selection of populations, in addition to comparisons with Italian and US DRIs, it may be valuable to add comparisons with the nutritional standards of other countries or international organizations.

We thank very much Rev#3 for this suggestion.

We compared the intakes obtained with the VegPlate also with Canada and UK DRIs (see table 4)

  1. In discussing the effects of low-fat vegan diets on various diseases, the authors could explore the interconnections between the mechanisms of action in greater depth, rather than merely listing the results of individual studies.

We thank Rev#3 for this suggestion. We improved with more details the following part (lines 224-253):

The abundance of fiber, water, and the limited content of fats of unprocessed plant foods are responsible of their low-calorie density, reducing the risk of excessive energy intakes. Thermic effect of foods increased after the consumption of large intakes of carbohydrates and low intakes of fats [28], and plant food favorably influenced the regulation of appetite and the intake of food and energy, by regulating the secretion of gastrointestinal hormones [29,30]. Providing the body with molecules with anti-oxidant proprieties, like biocompounds naturally contained in plants - especially when grown in harsh environments - can protect the body from the damage caused by oxidative stress [31]: the consumption of whole plant foods, rich in antioxidants, has been associated with a reduction in the risk of major chronic diseases [32]. Accordingly, it has been reported that a vegetarian diet can reduce the levels of oxidative stress markers and the oxidant-antioxidant balance compared to an omnivore diet [33,34]. Moreover, a plant-based diet can reduce iron accumulation in the tissues [35], which can be related with multiple health outcomes [36]. Plant foods can favor the growth of beneficial bacteria, and of a greater richness and diversity of gut microbiota: it has been reported that vegetarians have a higher Prevotella/Bacteroides ratio and a lower Firmicutes/Bacteroidetes ratio [37]. Short-chain fatty acids (acetate, propionate, and butyrate), produced during the bacterial fermentation of dietary fiber, elicit beneficial effects on immunity, inflammation, lipid and glucose metabolism, gut-barrier, and blood-brain barrier integrity [38–41]. Plasma and urinary TMAO concentration (trimethylamine N-oxide, a marker of cardiovascular disease risk) are increased by dietary protein, particularly of animal origin, and reduced in subjects following plant-based diets [42]. Low-grade chronic inflammation represents a common underlying factor in chronic diseases [43–47]. Lower levels of inflammatory biomarkers, mainly hsCRP, have been reported in vegetarians [48–51], suggesting that the reduction of their circulating levels in subjects following a plant-based diet could reduce the risk of chronic diseases. Accordingly, in a clinical trial performed on subjects affected by coronary artery disease, a -32% significant reduction of hsCRP (high-sensitivity C-Reactive Protein, a marker of risk for major adverse cardiovascular outcomes in coronary artery disease) was observed after 8 weeks of vegan diet, in comparison with the diet proposed by American Heart Association [52].

  1. It is recommended that references be primarily from the last five years. Please carefully format references according to the formatting requirements of the Foods, many articles are missing page numbers.

We thank Rev#3 for this comment. We selected the most recent papers, but some not recent papers represent a milestone of the research in this field, and authors trust that it’s worth to propose their results. We corrected the formatting of the references where page numbers were available (for some journal the “cite” function in Pubmed quotes only the first page).

Round 2

Reviewer 3 Report

Comments and Suggestions for Authors

The authors have carefully and revised the manuscript according to the suggestions and the quality of the manuscript has been improved. However, before the article is accepted, I suggest minor revisions.

1. Figure 1 needs landscaping.

2. References have a lot of missing page numbers, as well as attention to italics.

3. The presentation of Table 1 is strange, can it be redesigned and focused.

4. Suggest rewriting the conclusion to highlight the significance and HIGHLIGHT of the study. it seems that some of the content is repeated a lot from the abstract.

Author Response

The authors have carefully and revised the manuscript according to the suggestions and the quality of the manuscript has been improved. However, before the article is accepted, I suggest minor revisions.

Authors sincerely thank Rev#3 for the attention and the further suggestion

1.Figure 1 needs landscaping.

Thank you, the figure has been landscaped

  1. References have a lot of missing page numbers, as well as attention to italics.

We thank very much Rev#3 for his/her attention.

We corrected some typos generated by Zotero.

Conversely, for some references the data are correctly indicated, here some examples from the “Cite” function of PubMed:

Ref 8: Baroni L, Rizzo G, Galchenko AV, Zavoli M, Serventi L, Battino M. Health Benefits of Vegetarian Diets: An Insight into the Main Topics. Foods. 2024 Jul 29;13(15):2398. doi: 10.3390/foods13152398. PMID: 39123589; PMCID: PMC11311397.

Ref 9: Landry MJ, Senkus KE, Mangels AR, Guest NS, Pawlak R, Raj S, Handu D, Rozga M. Vegetarian dietary patterns and cardiovascular risk factors and disease prevention: An umbrella review of systematic reviews. Am J Prev Cardiol. 2024 Sep 28;20:100868. doi: 10.1016/j.ajpc.2024.100868. PMID: 39430429; PMCID: PMC11489049.

Ref 10: Capodici A, Mocciaro G, Gori D, Landry MJ, Masini A, Sanmarchi F, Fiore M, Coa AA, Castagna G, Gardner CD, Guaraldi F. Cardiovascular health and cancer risk associated with plant based diets: An umbrella review. PLoS One. 2024 May 15;19(5):e0300711. doi: 10.1371/journal.pone.0300711. PMID: 38748667; PMCID: PMC11095673.

3.The presentation of Table 1 is strange, can it be redesigned and focused.

We thank Rev#3 for pointing out this criticism. Table 1 has been redesigned and focused

  1. Suggest rewriting the conclusion to highlight the significance and HIGHLIGHT of the study. it seems that some of the content is repeated a lot from the abstract.

We thank Rev#3 for this suggestion. We modified the Conclusion as follows:

The benefits of a low-fat diet combined with a vegan plan have been explored in clinical trials. This promising approach appears to be beneficial to health, is consistent with the planetary goals of reducing the environmental impact of diets and, with good and regular support, adherence rates to low-fat vegan diets are more than solid. Nevertheless, to date no food guide for planning low-fat vegan diets was available for health professional. The VegPlate_Low-Fat (VP_LF) aims to represent a new proposal for an easy-to-use vegetarian food guide for the planning of low-fat vegan diets, to be used by health professionals in their daily activity. Despite further research to confirm and eventually enhance the application of low-fat vegan diets in clinical practice is warranted, we hope that this new proposal can represent a useful tool for nutrition professionals, and can be considered a complementary first line, inexpensive approach to cardiometabolic diseases. We expect that further intervention studies performed with real diets based on the VP_LF method can confirm its effectiveness for joining nutrition adequacy with therapeutic advantages. It would also be useful to compare their effectiveness with that of omnivorous diet with similar amounts of fats.